# Participant characteristics in the prevention of gestational diabetes as evidence for precision medicine: a systematic review and meta-analysis

Siew Lim [1✉], Wubet Worku Takele[1,204], Kimberly K. Vesco[2,204], Leanne M. Redman[3,204], Wesley Hannah [4,5,204], Maxine P. Bonham[6], Mingling Chen [7], Sian C. Chivers [8], Andrea J, Fawcett[9,10], Jessica A. Grieger[11], Nahal Habibi[11], Gloria K. W. Leung[6], Kai Liu[6], Eskedar Getie Mekonnen[12], Maleesa Pathirana[11], Alejandra Quinteros [11], Rachael Taylor[13], Gebresilasea G. Ukke[1], Shao J. Zhou[14,15], ADA/EASD PMDI* & Jami Josefson[16✉]

## Abstract

**Background** Precision prevention involves using the unique characteristics of a particular group to determine their responses to preventive interventions. This study aimed to systematically evaluate the participant characteristics associated with responses to interventions in gestational diabetes mellitus (GDM) prevention.

**Methods** We searched MEDLINE, EMBASE, and Pubmed to identify lifestyle (diet, physical activity, or both), metformin, myoinositol/inositol and probiotics interventions of GDM prevention published up to May 24, 2022.

**Results** From 10347 studies, 116 studies ($n = 40940$ women) are included. Physical activity results in greater GDM reduction in participants with a normal body mass index (BMI) at baseline compared to obese BMI (risk ratio, 95% confidence interval: 0.06 [0.03, 0.14] vs 0.68 [0.26, 1.60]). Combined diet and physical activity interventions result in greater GDM reduction in participants without polycystic ovary syndrome (PCOS) than those with PCOS (0.62 [0.47, 0.82] vs 1.12 [0.78–1.61]) and in those without a history of GDM than those with unspecified GDM history (0.62 [0.47, 0.81] vs 0.85 [0.76, 0.95]). Metformin interventions are more effective in participants with PCOS than those with unspecified status (0.38 [0.19, 0.74] vs 0.59 [0.25, 1.43]), or when commenced preconception than during pregnancy (0.21 [0.11, 0.40] vs 1.15 [0.86–1.55]). Parity, history of having a large-for-gestational-age infant or family history of diabetes have no effect on intervention responses.

**Conclusions** GDM prevention through metformin or lifestyle differs according to some individual characteristics. Future research should include trials commencing preconception and provide results disaggregated by a priori defined participant characteristics including social and environmental factors, clinical traits, and other novel risk factors to predict GDM prevention through interventions.

## Plain language summary

An individual's characteristics, such as medical, biochemical, social, and behavioural may affect their response to interventions aimed at preventing gestational diabetes, which occurs during pregnancy. Here, we evaluated the published literature on interventions such as diet, lifestyle, drug treatment and nutritional supplement and looked at which individual participant characteristics were associated with response to these interventions. Certain participant characteristics were associated with greater prevention of gestational diabetes through particular treatments. Some interventions were more effective when started prior to conception. Future studies should consider individual characteristics when assessing the effects of preventative measures.

A full list of author affiliations appears at the end of the paper.

Gestational diabetes mellitus (GDM) is characterized by glucose intolerance first identified during pregnancy and is associated with perinatal and long-term adverse health outcomes in both the pregnant individual and the offspring. The physiologic reduction in insulin sensitivity during pregnancy is the hallmark metabolic feature that leads to the onset of glucose intolerance and GDM in predisposed individuals[1]. Established risk factors for GDM include previous GDM, advanced maternal age, parity, overweight/obesity, and family history of diabetes[2,3]. GDM increases perinatal complications including preeclampsia, operative deliveries, stillbirth, neonatal risks of large-for-gestational age, hypoglycemia, and respiratory distress syndrome[4]. GDM confers an increased lifetime risk of type 2 diabetes mellitus for both mother and offspring[5,6]. GDM rates vary considerably, with geographic differences and varying diagnostic criteria accounting for the 1–30% incidence[7]. Nonetheless, rates of GDM are increasing across all populations[8,9], commensurate to worldwide increasing rates of overweight and obesity.

Prevention of GDM involves reducing hyperglycemia and insulin resistance, factors that are also highly correlated with obesity[10,11]. Weight reduction prior to pregnancy and prevention of excessive gestational weight gain (GWG) are important features of diabetes prevention[12,13]. Insulin resistance is affected by a number of factors: weight, lifestyle, physical activity, dietary intake and supplement use. Several meta-analyses of randomized controlled trials (RCTs) investigating lifestyle interventions have reported on diet and physical activity interventions, metformin, and supplements as either primary GDM prevention strategies or secondary prevention strategies for trials targeting weight management and/or reduction as a primary outcome. Results of these meta-analyses have not been unanimous in the reporting of findings suggesting heterogeneity in the intervention response, perhaps due to the characteristics of the study population, and/or the timing and type of intervention[14–17].

Individual characteristics such as clinical, psychosocial and biochemical factors may influence the effectiveness of interventions in preventing GDM. The prevention of GDM results from an interaction between behavioural factors, such as the ability to adhere to the intervention, and physiological factors, such as the biological responsiveness towards reducing insulin resistance. Hence, clinical traits, such as overweight/obesity, age, history of GDM or polycystic ovary syndrome (PCOS), along with social determinants of health, for example, socioeconomic status, cultural background, race or ethnicity, are potential sources of heterogeneity of the intervention effect[18]. These clinical, biochemical, social and environmental traits could affect GDM prevention through behavioural or physiological pathways, or both. Given that interventions to prevent GDM are unlikely to be effective for individuals as a 'one-size-fits-all' approach, there is a need to elucidate the most effective mode of prevention for each population. To date, there has not been a comprehensive meta-analysis of GDM prevention, accounting for participant characteristics to inform precision medicine.

The field of precision medicine recognizes that examining the heterogeneity of individual responses to intervention is important for optimizing health-enhancing interventions and minimizing exposure to specific risk factors, to delay or prevent the onset of a given disease[18,19]. The Precision Medicine in Diabetes Initiative (PMDI) was established in 2018 by the American Diabetes Association (ADA) in partnership with the European Association for the Study of Diabetes (EASD). The ADA/EASD PMDI includes global thought leaders in precision diabetes medicine who are working to address the burgeoning need for better diabetes prevention and care through precision medicine[18]. This review is written on behalf of the ADA/EASD PMDI as part of a comprehensive evidence evaluation in support of the 2nd International Consensus Report on Precision Diabetes Medicine[20]. To inform a precision medicine approach to diabetes prevention, the primary focus of this review was to assess the contribution of various participant characteristics to the effectiveness of interventions for GDM prevention. To this end, this systematic review and meta-analysis examined the effectiveness of interventions employing lifestyle modification, metformin, or dietary supplements within the preconception, pregnant and postpartum/interconception periods for reducing the risk of developing GDM. We find that certain participant characteristics such as BMI, having polycystic ovary syndrome or history of GDM or being in the preconception phase may determine responses to particular interventions.

## Methods

This systematic review and meta-analysis was conducted according to the Preferred Reporting Items for Systematic Reviews and Meta-Analyses (PRISMA) Statement[21]. The protocol was registered in the PROSPERO International Prospective Register of Systematic Reviews (CRD42022320513).

**Search Strategy.** A comprehensive search strategy was developed by a professional medical librarian (AF) in consultation with the authors (SL, LR, KV, JJ). The search strategy included keywords and Medical Subject Headings (MeSH), as shown in Supplementary Table 1. We searched the following databases: Embase (Elsevier), Ovid Medline, and PubMed from the inception of the database to May 24, 2022. Results were limited to studies in human and in English-language. No limit was placed on publication date. Endnote (Clarivate) was used to compile records and remove duplicates. Covidence (Veritas Health Innovation, Melbourne, Australia) was then used for title/abstract screening and full text review. Hand-searches including the reference list of related reviews were also examined for additional eligible trials.

**Selection criteria.** Randomised and non-randomized controlled trials (RCTs and non-RCTs) investigating the effects of lifestyle (diet, exercise, or both), metformin, or dietary supplements (fish oil, myoinositol/inositol, probiotics) on prevention of GDM in women of childbearing age (including preconception cohorts) were included (Supplementary Table 2). Control conditions included usual care or minimal intervention (no more than a single intervention session in the case of diet and exercise interventions). Studies without a control group (usual care or placebo), those that did not report GDM, observational studies, editorials, commentaries, conference abstracts, reviews, meta-analyses and study protocols were excluded. Titles and abstracts were evaluated independently and in duplicate to identify articles for full-text review. Full-text review was conducted independently and in duplicate with reasons for exclusion recorded. Discrepancies were resolved by consensus by two or more authors.

**Data extraction.** Data were extracted using an extraction template developed for this study with GDM as the primary outcome. Study characteristics (authors, year of publication, country, setting, sample size, design, diagnostic criteria, diagnosis time point, intervention commencement, and outcome of interest), participant characteristics (age, race/ethnicity, BMI, education status, employment status, parity, prior GDM, smoking status and other medical history), intervention type (diet, physical activity, diet and physical activity, metformin, types of dietary supplement), and outcome of intervention (GDM incidence) were extracted. Study characteristics were determined based on known GDM risk factors and other relevant factors identified by the precision

medicine report[19]. Authors were contacted for missing information. One author conducted the data extraction, and a second author conducted a 10% sub-sample data extraction to establish reliability. An agreement of 89% was achieved between the two authors with discussion to resolve discrepancies.

**Quality assessment.** Quality of the included studies was critically appraised using a relevant tool for each study design. The Revised Cochrane Risk of Bias Tool for Randomized Trials (RoB 2.0)[22] was used for RCTs to assess bias arising from the randomization process, deviations from the protocol, missing data, measurement of the outcome and selective reporting. The ROBINS-I tool was used for non-RCTs to assess bias from confounding, participant selection, classification of interventions, missing data, deviations from intended interventions, measurement of outcomes, and selection of reported results[23]. Two reviewers independently conducted the methodological quality and bias assessment for individual studies. Differences were resolved by consensus.

The GRADE process (Grading of Recommendations Assessment, Development and Evaluation), which rates the quality of evidence from a study in a systematic approach was conducted for the primary outcome of GDM[24]. Risk of bias, along with consistency, directness, precision and publication bias were considered for GRADE appraisal to determine the quality of evidence.

**Statistical analysis.** The outcome was the incidence of GDM. Data were pooled and GDM incidence was expressed as risk ratios (RR) with 95% confidence intervals (CI). Heterogeneity between studies was assessed by the $I^2$ test where $I^2 > 50\%$ indicated substantial heterogeneity. Potential sources of heterogeneity by participant characteristics (e.g. obesity, age, PCOS) were explored through subgroup analyses and meta-regression, as conducted in other systematic reviews and meta-analyses[25,26]. Significant ($p < 0.05$) Egger's test and funnel plot (asymmetry) was used to declare publication bias. Estimates (RR) were pooled using random-effects model with the DerSimonian and Laird estimator[27]. Subgroup analyses were conducted if there was at least one trial present in at least two comparative subgroups. Sensitivity analyses were conducted by excluding non-randomized controlled trials. $P < 0.05$ was taken as the level of statistical significance. All analyses were conducted in Stata Version 17 (STATA Corporation, College Station, Texas, USA).

**Reporting summary.** Further information on research design is available in the Nature Portfolio Reporting Summary linked to this article.

## Results

We screened 10,347 records for eligibility and 434 records were reviewed as full texts (Fig. 1). Overall, 130 articles were deemed eligible representing 116 unique studies (117 comparisons due to multiple intervention arms in Luoto et al) and were included in the meta-analysis. Reasons for exclusion included lack of an appropriate control group where the only difference between the treatment and control group was with or without the interventions of interest, no GDM outcome, or active intervention in the control group (Fig. 1).

**Study characteristics.** A summary of the characteristics of included studies are shown in Supplementary Data 1. Studies were published from 1997 to 2022. Sample sizes ranged from 31 to 4631. Of the included studies, 92 (79%) involved lifestyle (diet, physical activity, or both), 13 (11%) involved metformin, and 12 (10%) involved dietary supplement interventions. One study included a comparison of lifestyle, probiotics with diet and

control[28] and was included as both a lifestyle and dietary supplement study in the meta-analysis. Types of lifestyle intervention included diet only ($n = 17$), physical activity only ($n = 19$) or a combination of diet and physical activity ($n = 59$). The types of dietary supplement interventions included myoinositol/inositol ($n = 7$), probiotics-only ($n = 4$), probiotics coupled with diet ($n = 1$), probiotics with fish oil ($n = 1$) and fish-oil only ($n = 1$). Interventions commenced from preconception to 26 weeks gestation.

The definition of the participant characteristics is shown in Supplementary Table 3. Detailed description of the participant characteristics of the included studies is shown in Supplementary Table 4. One hundred and five studies commenced the intervention during pregnancy, seven studies reported commencing the intervention prior to pregnancy while four studies did not provide information on pregnancy status at recruitment. Two studies were conducted in women who were nulliparous. A number of studies included only participants with certain medical conditions or medical history: overweight (BMI 25–29.9 kg/m$^2$) or obesity (BMI > 29.9 kg/m$^2$) ($n = 57$), PCOS ($n = 9$), prediabetes ($n = 1$) or family history of diabetes ($n = 2$); whereas some studies excluded participants with certain medical conditions or medical history: hypertension ($n = 27$), prediabetes ($n = 52$), PCOS ($n = 3$), history of stillbirth ($n = 2$), family history of diabetes ($n = 2$), previous macrosomia infant ($n = 3$), past history of GDM ($n = 20$), hypertensive disorders of pregnancy ($n = 4$), history of cardiovascular disease ($n = 14$), smoking ($n = 11$). The mean age of the participants ranged from 25 to 34 years, and mean BMI at baseline ranged from 21 to 39 kg/m$^2$. Of the included studies, 25 (22%) had mostly participants with tertiary education (as defined in Supplementary Table 3) and 25 (22%) had mostly participants in employment (Supplementary Table 3). Of the 54 studies which reported race, 18 studies were predominantly conducted among White participants, 10 were predominantly conducted among non-White participants, and 26 among mixed populations.

Each included study reported GDM as a primary or secondary outcome (Supplementary Data 1). The criteria used for GDM diagnosis varied across the studies and included one-step (most commonly a single 75-gram, 2-h oral glucose tolerance test) and two-step methods (commonly the 50 gram one-hour oral glucose challenge test, followed by a 2 or 3-h oral glucose tolerance test if the oral glucose challenge test was abnormal). The most frequently reported diagnostic criteria ($n = 37$) were those of the International Association of the Diabetes and Pregnancy Study Groups (IADPSG), the World Health Organization (WHO) in 1999 (prior to WHO adopting those of the IADPSG) ($n = 9$), Carpenter & Coustan (C&C, $n = 7$), and National Diabetes Data Group (NDDG, $n = 6$). The method of GDM diagnosis was not reported by 34 studies and 13 studies used a method that could not be categorized by one set of diagnostic criteria. Of the 37 studies using IADPSG criteria, 22 tested diet and physical activity interventions, 9 supplements, 3 metformin, 2 diet, and 1 physical activity. Within those studies that used IADPSG, 26 identified GDM as a primary outcome. Of the 7 studies initiated in the preconception period (2 diet, 1 diet+physical activity, 4 metformin), GDM was the primary outcome for 4, 2 of which were non-randomized clinical studies of metformin use initiated prior to pregnancy targeting women with PCOS and the only 2 that used the same criteria for diagnosing GDM (NDDG).

**Meta-analysis.** Meta-analysis of all lifestyle interventions (diet only, physical activity only, and combined diet+physical activity) showed a significant reduction in the risk of GDM with moderate heterogeneity (RR 0.78, 95%CI 0.72, 0.85, $I^2 = 45$). (Table 1).

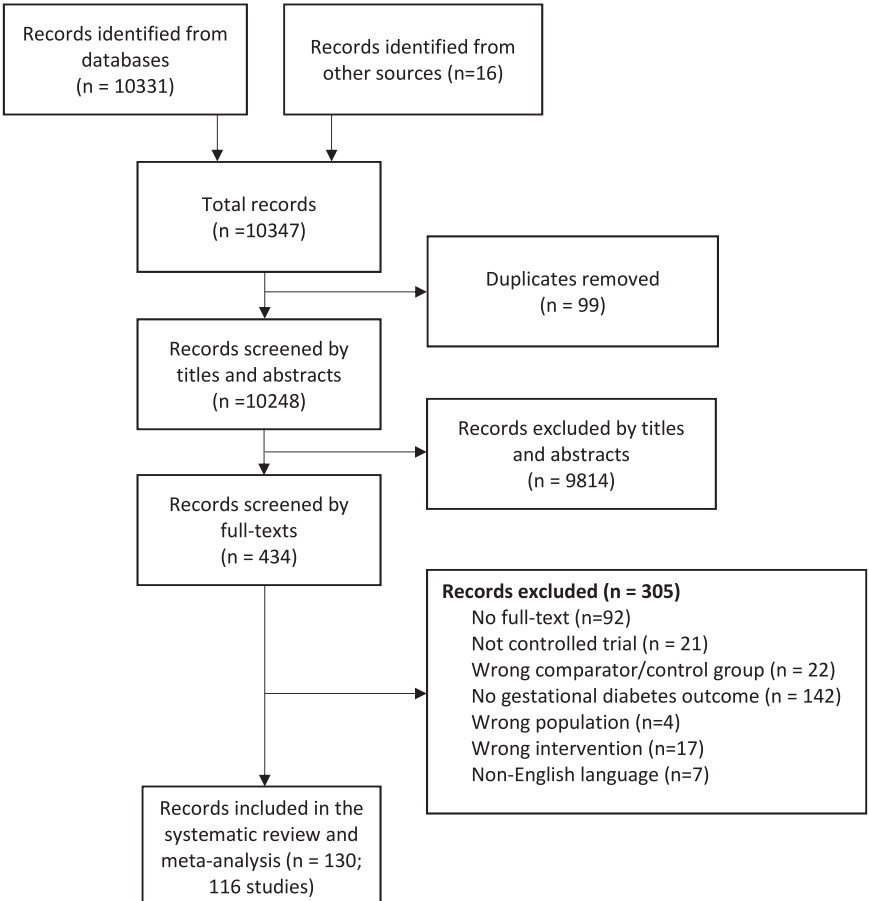

**Fig. 1 PRISMA flow diagram.** Flowchart demonstrating the process of the identifying the papers included in this review. This maps out the number of records identified, included and excluded at each stage of assessment. Reasons for full text exclusions are provided.

*Diet-only*. In the random effects model, diet-only interventions showed a significant reduction in the risk of GDM (RR 0.78, 95% CI 0.65, 0.94, 17 studies, $I^2 = 45.6\%$, moderate-quality evidence) with moderate heterogeneity.

Risk differences in the subgroup and meta-regression analyses by participant characteristics such as sociodemographic (e.g. educational status) and medical history (e.g. prediabetes and hypertension) was not observed (Supplementary Data 2 and 3).

*Physical activity-only*. Meta-analysis of physical activity-only interventions showed a significant reduction in the risk of GDM (RR 0.70, 95%CI 0.57, 0.87, 19 studies, $I^2 = 28.6\%$, moderate-quality evidence) with moderate heterogeneity.

Subgroup analyses showed that physical activity-only interventions resulted in greater reduction in risk for GDM in studies involving women with normal BMI compared with other BMI groups, and in interventions commencing before 12 gestation weeks (Supplementary Data 4).

*Diet and physical activity*. Meta-analysis of interventions with both diet and physical components showed a significant reduction in the risk of GDM (RR 0.82, 95%CI 0.73, 0.91, 59 studies, $I^2 = 47\%$, low-quality evidence) with significant heterogeneity.

Subgroup analyses showed that diet and physical activity interventions were effective in studies involving women with overweight or obesity, but not in studies involving women with normal weight (Supplementary Data 5). Diet and physical activity interventions were more effective in reducing GDM in studies involving women without PCOS compared to those with PCOS, and in studies involving women without a history of GDM compared with those to unspecified history of GDM (Supplementary Data 5). Meta-regression showed that diet and physical activity interventions had greater reduction in GDM with increasing age (Supplementary Data 3).

*Metformin*. Meta-analysis of all metformin interventions showed a significant reduction in the risk of GDM (RR 0.66, 95%CI 0.47, 0.93, 13 studies, $I^2 = 73\%$, very low-quality evidence) with significant heterogeneity.

Subgroup analyses showed that metformin interventions were more effective when commenced preconception compared with during pregnancy (Supplementary Data 6). Metformin interventions were also more effective in reducing GDM in studies involving women with PCOS than those with unspecified status, and less effective in studies involving women without a history of GDM than those with unspecified history of GDM (Supplementary Data 6). Meta-regression showed that metformin interventions were more effective in reducing GDM with increasing age or higher fasting blood glucose at baseline (Supplementary Data 3).

*Myoinositol/Inositol*. Meta-analysis of myoinositol/inositol interventions showed a significant reduction in the risk of GDM (RR 0.39, 95%CI 0.23, 0.66, 7 studies, $I^2 = 79\%$, very low-quality evidence) with significant heterogeneity.

Subgroup analyses showed that myoinositol/inositol interventions were more effective in reducing GDM in studies involving mostly White women compared with women from various ethnic backgrounds (Supplementary Data 7).

**Table 1 Meta-analysis of the effect of diet, physical activity, metformin or dietary supplement on gestational diabetes prevention.**

| Intervention type | Number of studies | Number of participants | Risk Ratio (95% CI) | $I^2$ (%) | Certainty by GRADE | Downgrade explanations |
|---|---|---|---|---|---|---|
| Diet | 17 | 7509 | 0.78 (0.65–0.94) | 45.6 | Moderate | Most studies have Some Concerns or High risk of bias |
| Physical activity | 19 | 6701 | 0.70 (0.57–0.87) | 28.6 | Moderate | Most studies have Some Concerns or High risk of bias |
| Diet and physical activity | 59 | 22,026 | 0.82 (0.73–0.91) | 47 | Low | Most studies have Some Concerns or a High risk of bias; High levels of heterogeneity |
| Metformin | 13 | 3120 | 0.66 (0.47–0.93) | 73 | Very low | Most studies have Some Concerns or High risk of bias; High levels of heterogeneity; Egger test suggests significant publication bias |
| Myoinositol | 7 | 1313 | 0.39 (0.23–0.66) | 79 | Very low | Most studies have Some Concerns or High risk of bias; Very high levels of heterogeneity |
| Probiotics | 5 | 1105 | 0.87 (0.52–1.46) | 73.2 | Very low | Most studies have Some Concerns or High risk of bias; Very high levels of heterogeneity; Pooled CI crosses 1.0 suggesting imprecision |

*Probiotics.* Meta-analysis of probiotics interventions showed no reduction in the risk of GDM (RR 0.88, 95%CI 0.52, 1.47, 5 studies, $I^2 = 74\%$, very low-quality evidence) with significant heterogeneity. Of these, one study with diet plus probiotics resulted in significant reduction in GDM (RR 0.36, 95%CI 0.18,0.72)[28], however four studies with probiotics-only[29–32] and one study with fish oil-plus probiotics[32] did not reduce GDM (RR ranging from 0.59 to 1.5).

Subgroup analyses and meta-regression were not conducted in probiotics interventions due to small numbers in each type of intervention.

*Fish oil.* The single fish oil intervention study found did not reduce risk for GDM (RR 1.09, 95%CI 0.64, 1.85)[32].

**Sensitivity analysis.** After excluding two non-RCT studies, diet-only interventions remained significant in preventing GDM (RR 0.75; 95% CI; 0.64, 0.88; $I^2 = 23\%$). After excluding six non-RCT studies, combined diet and physical activity interventions remained significant in reducing GDM (RR 0.83; 95% CI; 0.74, 0.93; $I^2 = 64.8\%$). After excluding five non-RCT studies, the effect of metformin on reducing the incidence of GDM was no longer significant (RR 1.05, 95% CI 0.89, 1.23; $I^2 = 45.1\%$).

Sensitivity analysis was not conducted on physical activity-only, probiotics and myoinositol/inositol studies as they all were RCTs.

**Assessment of bias and quality of evidence.** Some concerns or high risk of bias were found in 12 (80%) of diet-only, 16 (94%) physical activity-only, 46 (78%) of diet and physical activity, 10 (77%) of metformin, 5 (71%) in myoinositol/inositol, 3 (60%) in probiotics (Supplementary Table 5). Most studies assigned a high risk of bias had insufficient or non-blinding of participants as the main reason, which for lifestyle intervention is essentially not possible.

The quality of evidence rated by the GRADE approach found that the overall quality for diet-only or physical activity-only interventions were moderate, downgraded mainly due to most studies contributing to the outcome having high or some concerns in risk of bias. (Supplementary Table 5). The quality

of evidence for diet and physical activity interventions were low, due to risk of bias and inconsistency. The quality of evidence for metformin interventions was very low, due to risk of bias, inconsistency and publication bias. The quality of evidence for myoinositol/inositol interventions was very low, due to risk of bias and inconsistency. The quality of evidence for probiotics interventions was very low due to risk of bias, inconsistency and imprecision.

**Publication bias.** Funnel plot and Egger's test suggested the presence of small studies publication bias for metformin studies (P = 0.001) (Supplementary Fig. 1).

No significant publication bias was detected for studies on physical activity-only, diet-only, diet and physical activity, or probiotics (Supplementary Figs. 2–6).

**Discussion**

This study aimed to determine the effect of participant characteristics in interventions for GDM prevention. Our analyses showed that lifestyle interventions, metformin and myoinositol/inositol reduced the risk of GDM. For physical activity-only interventions, greater risk reduction for GDM was seen in studies involving women with normal BMI. Combined diet and physical activity interventions were more effective in GDM reduction in those with overweight or obesity, without PCOS, without history of GDM and with increasing age. Metformin interventions were more effective in GDM reduction in women with a history of PCOS and with increasing age and fasting blood glucose. Metformin or physical activity-only interventions were more effective when commenced preconception or in early gestation (before 12 gestation weeks).

Diet and physical activity interventions were more effective in lowering the risk for GDM among women without a history of GDM. This may be because women with a prior history of GDM have impaired beta-cell compensatory response during pregnancy and persistent or ongoing decline in insulin sensitivity post-GDM pregnancy[11]. Lifestyle modification alone may have limited ability to overcome these impairments in glycemic control in these individuals. In addition, lower adherence to a healthy diet, as lower dietary quality, has been observed among women with a history of

GDM compared with women without a history of GDM, which may have also contributed to this finding[33]. Similarly, we also found that lifestyle interventions were more effective in lowering GDM risks in women without a history of PCOS. Like GDM, PCOS is also associated with increased insulin resistance[34]. In addition to the physiological challenges of insulin resistance, women with PCOS may also face further challenges with adhering to a healthy lifestyle which ranges from physiological barriers such as alteration in gut hormone regulation to psychological barriers such as a high prevalence of disordered eating in this population[35]. Further, we found that physical activity interventions were only effective in individuals with normal BMI but not in those with obese BMI. Similar observations have been reported in other meta-analyses of GDM prevention in individuals with excess body weight[36,37]. As obesity is associated with increased insulin resistance, it is possible that physical activity interventions alone could not reduce insulin resistance sufficiently for the prevention of GDM in these individuals. Further research is needed to determine if more intensive lifestyle intervention or additional co-intervention such as metformin or supplementation is needed to prevent GDM in women with conditions of high insulin resistance including prior GDM, PCOS and/or excess body weight. This is important so as not to provide unfounded expectations on the benefit of lifestyle on GDM prevention for certain groups. It is also of particular pertinence to these specific populations outlined, given the stigma associated with obesity and diabetes, resulting from the perception that these health outcomes are caused by personal failures[38,39].

To date, systematic reviews are inconsistent in the observed effect of metformin on GDM prevention[16,40]. Heterogeneity in participant characteristics across studies, which undermines the power to detect a significant effect within a small number of studies, may contribute to this inconsistency[16,40]. Past studies conducted in select populations with homogeneous characteristics, such as studies in women with PCOS, have found a consistent benefit in GDM prevention with metformin[41,42]. By increasing the number of included metformin trials from 3 in the previous meta-analysis[16] to 13, the current review revealed a significant reduction in GDM with metformin. To further explore the sources of heterogeneity by participant characteristics, our meta-regression additionally found that metformin is more effective in studies involving women with an older mean age at baseline, or in women with higher baseline fasting blood glucose. Increasing age is associated with greater insulin resistance, while higher baseline fasting glucose indicates early signs of failure of beta-cell compensatory response in insulin production[43]. The greater benefit of metformin in women with increased insulin resistance is in line with the known mechanisms of metformin, which is to reduce glucose production in the liver, improve peripheral glucose uptake and increase insulin sensitivity[44]. Our findings suggest that metformin may be the intervention of choice for preventing GDM in populations at high risk of insulin resistance, including in women with advanced age, higher fasting blood glucose, history of GDM or PCOS, along with healthy antenatal lifestyle advice. However, we found that the quality of evidence for metformin in preventing GDM was very low, thus further high-quality RCTs are needed to confirm these findings.

In this review, we identified seven preconception interventions that met our inclusion criteria. Despite this small number, metformin or physical activity-only interventions were more effective in lowering the risk for GDM when commenced preconception or in the first trimester of pregnancy. Our conclusions are similar to those of a previous meta-analysis[15]. Earlier initiation of interventions results in a greater duration of intervention exposure prior to GDM diagnosis, with the benefit of preconception commencement providing an opportunity to optimize insulin sensitivity prior to the onset of pregnancy-induced insulin resistance. A preconception intervention is in line with the concept of GDM arising as a result of a chronic metabolic condition that antedates pregnancy. Although GDM may be the first recognition of impaired fasting glucose or glucose intolerance, data suggest that women who develop GDM are already on a trajectory of increased cardiometabolic risk prior to pregnancy[45]. Future research should focus on providing preconception GDM prevention in those at risk of developing GDM. The use of preconception risk prediction models for GDM may help identify the populations who will optimally benefit from early initiation interventions[46].

The strength of this review includes a comprehensive assessment of the impact of participant characteristics on the effectiveness of a broad range of interventions for the prevention of GDM, with a goal to identify populations that could optimally benefit from each intervention type. Although most studies were conducted in mixed populations and did not report outcomes according to subgroup characteristics, we coded the participant subgroups according to the inclusion and exclusion criteria (e.g. if the study included only women with obesity) to allow for group comparisons. This review also has several limitations. Participant characteristics that are known risk factors for GDM, such as parity, seldom feature in the inclusion or exclusion criteria of relevant studies, yielding a low number of subgroups available for comparison. This could be partly mitigated in individual level meta-analysis, if this information were collected at an individual level by the original studies. Significant heterogeneity remained in some subgroups, suggesting that other confounding factors may have contributed to the effect sizes. One of the confounding effects that are difficult to quantify is the changing GDM diagnostic criteria over the years which may have also contributed to the heterogeneity observed in the analyses. The findings on lifestyle or metformin on GDM incidence may also be limited by possible publication bias. The certainty of the evidence was very low for metformin, myoinositol/inositol and probiotics and low for diet and physical activity combined, and should be interpreted with caution.

## Conclusion

Lifestyle, metformin and myoinositol/inositol interventions reduce the risk of GDM. Lower GDM risks were seen when the intervention commenced preconception or in the first trimester of pregnancy. Diet and physical activity interventions may be associated with a greater reduction in GDM risks in women with older age or without a history of GDM or PCOS, while metformin may be more effective in preventing GDM in women with older age, having higher fasting blood glucose or with PCOS. However, these results should be interpreted with caution due to limited reporting of intervention outcomes by participant characteristics in the individual studies. Given the potentially greater effectiveness of lifestyle and metformin interventions in individuals, future research on tailored recommendations in precision GDM prevention, replacing the current 'one-size-fits-all' approach, is needed. To advance knowledge in precision prevention, future research should include trials commencing in the preconception period and provide results disaggregated by a priori defined participant characteristics, including social and environmental factors, clinical traits, and other novel risk factors.

## Data availability

All data generated or analysed during this study are included in this published article and its Supplementary Information files. The list of included studies is available in Supplementary Data 1.

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

## Acknowledgements

The ADA/EASD Precision Diabetes Medicine Initiative, within which this work was conducted, has received the following support: The Covidence license was funded by Lund University (Sweden) for which technical support was provided by Maria Björklund and Krister Aronsson (Faculty of Medicine Library, Lund University, Sweden). Administrative support was provided by Lund University (Malmö, Sweden), University of Chicago (IL, USA), and the American Diabetes Association (Washington D.C., USA). The Novo Nordisk Foundation (Hellerup, Denmark) provided grant support for in-person writing group meetings (PI: L Phillipson, University of Chicago, IL).SL is funded by the Australian National Health and Medical Research Council (NHMRC) Fellowship. JW is funded by NHMRC Ideas Grant. WT and MC are funded by the Australian Government Research Training Program Scholarship. GGU is funded by the Monash Graduate Scholarship and Monash International

Tuition Scholarship. SC is funded by the Sir George Alberti fellowship, Diabetes UK (21/0006277). LR is funded by the National Institute of Health (5R01DK124806). JJ is funded by the National Institute of Health (5R01DK118403). The Novo Nordisk Foundation (Hellerup, Denmark) provided grant support for in-person meetings (PI: L. Phillipson, University of Chicago, IL, USA).

## Author contributions

S.L., J.J., L.R., and K.V. conceptualised the research question. A.F. contributed to the search of the articles. S.L., M.C., J.G, N.H., L.R., J.J., K.V., W.T., K.L., G.U., and S.C. screened the articles. J.G., N.H., W.T., G.U., G.L., S.Z., R.T., M.P., K.L., M.B., A.Q., W.H., and E.M .extracted the data and appraised the studies. W.T. and S.L. coded the participant characteristics. W.T., G.U. and M.C. contributed to data analysis, all authors contributed to the manuscript.

## Competing interests

The authors declare no competing interests.

## Additional information

[1]Eastern Health Clinical School, Monash University, Melbourne, Victoria, Australia. [2]Kaiser Permanente Northwest, Kaiser Permanente Center for Health Research, Oakland, USA. [3]Pennington Biomedical Research Center, Baton Rouge, LA, USA. [4]Madras Diabetes Research Foundation Chennai, Chennai, India. [5]Deakin University, Melbourne, Australia. [6]Department of Nutrition and Dietetics, Monash University, Melbourne, Victoria, Australia. [7]Monash Centre for Health Research and Implementation, Monash University, Clayton, VIC, Australia. [8]Department of Women and Children's Health, King's College London, London, United Kingdom. [9]Ann & Robert H. Lurie Children's Hospital of Chicago, Chicago, IL, USA. [10]Department of Clinical & Organizational Development, University of Chicago, Chicago, IL, USA. [11]Adelaide Medical School, Faculty of Health and Medical Sciences, The University of Adelaide, Adelaide, Australia. [12]Global Health Institute, University of Antwerp, Antwerp, Belgium. [13]School of Health Sciences, University of Newcastle, Newcastle, Australia. [14]School of Agriculture, Food and Wine, University of Adelaide, Adelaide, Australia. [15]American Diabetes Association (ADA) and European Association for the Study of Diabetes (EASD) Precision Medicine in Diabetes Initiative (PMDI) led by Paul Franks, Malmo, Sweden. [16]Northwestern University/ Lurie Children's Hospital of Chicago, Chicago, USA. [204]These authors contributed equally: Wubet Worku Takele, Kimberly K. Vesco, Leanne M. Redman, Wesley Hannah. *A list of authors and their affiliations appears at the end of the paper. ✉email: siew.lim1@monash.edu; JJosefson@luriechildrens.org

## ADA/EASD PMDI

Deirdre K. Tobias[17,18], Jordi Merino[19,20,21], Abrar Ahmad[22], Catherine Aiken[23,24], Jamie L. Benham[25], Dhanasekaran Bodhini[26], Amy L. Clark[27], Kevin Colclough[28], Rosa Corcoy[29,30,31], Sara J. Cromer[20,32,33], Daisy Duan[34], Jamie L. Felton[35,36,37], Ellen C. Francis[38], Pieter Gillard[39], Véronique Gingras[40,41], Romy Gaillard[42], Eram Haider[43], Alice Hughes[28], Jennifer M. Ikle[44,45], Laura M. Jacobsen[46], Anna R. Kahkoska[47], Jarno L. T. Kettunen[48,49,50], Raymond J. Kreienkamp[20,21,32,51], Lee-Ling Lim[52,53,54], Jonna M. E. Männistö[55,56], Robert Massey[43], Niamh-Maire Mclennan[57], Rachel G. Miller[58], Mario Luca Morieri[59,60], Jasper Most[61], Rochelle N. Naylor[62], Bige Ozkan[63,64], Kashyap Amratlal Patel[28], Scott J. Pilla[65,66], Katsiaryna Prystupa[67,68], Sridharan Raghavan[69,70], Mary R. Rooney[63,71], Martin Schön[67,68,72], Zhila Semnani-Azad[18], Magdalena Sevilla-Gonzalez[32,33,73], Pernille Svalastoga[74,75], Wubet Worku Takele[1,204], Claudia Ha-ting Tam[54,76,77], Anne Cathrine B. Thuesen[19], Mustafa Tosur[78,79,80], Amelia S. Wallace[63,71], Caroline C. Wang[71], Jessie J. Wong[81], Jennifer M. Yamamoto[82], Katherine Young[28], Chloé Amouyal[83,84], Mette K. Andersen[19], Maxine P. Bonham[85], Mingling Chen[7], Feifei Cheng[86], Tinashe Chikowore[33,87,88,89], Sian C. Chivers[8], Christoffer Clemmensen[19], Dana Dabelea[90], Adem Y. Dawed[43], Aaron J. Deutsch[21,32,33], Laura T. Dickens[91], Linda A. DiMeglio[35,36,37,92], Monika Dudenhöffer-Pfeifer[22], Carmella Evans-Molina[35,36,37,93], María Mercè Fernández-Balsells[94,95], Hugo Fitipaldi[22], Stephanie L. Fitzpatrick[96], Stephen E. Gitelman[97], Mark O. Goodarzi[98,99], Jessica A. Grieger[11,100], Marta Guasch-Ferré[18,101],

Nahal Habibi[11,100], Torben Hansen[19], Chuiguo Huang[54,76], Arianna Harris-Kawano[35,36,37], Heba M. Ismail[35,36,37], Benjamin Hoag[102,103], Randi K. Johnson[104,105], Angus G. Jones[28,106], Robert W. Koivula[107], Aaron Leong[20,33,108], Gloria K. W. Leung[85], Ingrid M. Libman[109], Kai Liu[11], S. Alice Long[110], William L. Lowe Jr.[111], Robert W. Morton[112,113,114], Ayesha A. Motala[115], Suna Onengut-Gumuscu[116], James S. Pankow[117], Maleesa Pathirana[11,100], Sofia Pazmino[118], Dianna Perez[35,36,37], John R. Petrie[119], Camille E. Powe[20,32,33,120], Alejandra Quinteros[11], Rashmi Jain[121,122], Debashree Ray[71,123], Mathias Ried-Larsen[124,125], Zeb Saeed[126], Vanessa Santhakumar[17], Sarah Kanbour[65,127], Sudipa Sarkar[65], Gabriela S. F. Monaco[35,36,37], Denise M. Scholtens[128], Elizabeth Selvin[63,71], Wayne Huey-Herng Sheu[129,130,131], Cate Speake[132], Maggie A. Stanislawski[104], Nele Steenackers[118], Andrea K. Steck[133], Norbert Stefan[68,134,135], Julie Støy[136], Rachael Taylor[137], Sok Cin Tye[138,139], Gebresilasea Gendisha Ukke[1], Marzhan Urazbayeva[79,140], Bart Van der Schueren[118,141], Camille Vatier[142,143], John M. Wentworth[144,145,146], Wesley Hannah[147], Sara L. White[8,148], Gechang Yu[54,76], Yingchai Zhang[54,76], Shao J. Zhou[14,100], Jacques Beltrand[149,150], Michel Polak[149,150], Ingvild Aukrust[74,151], Elisa de Franco[28], Sarah E. Flanagan[28], Kristin A. Maloney[152], Andrew McGovern[28], Janne Molnes[74,151], Mariam Nakabuye[19], Pål Rasmus Njølstad[74,75], Hugo Pomares-Millan[22,153], Michele Provenzano[154], Cécile Saint-Martin[155], Cuilin Zhang[156,157], Yeyi Zhu[158,159], Sungyoung Auh[160], Russell de Souza[113,161], Andrea J. Fawcett[162,163], Chandra Gruber[164], Eskedar Getie Mekonnen[165,166], Emily Mixter[167], Diana Sherifali[113,168], Robert H. Eckel[169], John J. Nolan[170,171], Louis H. Philipson[167], Rebecca J. Brown[160], Liana K. Billings[172,173], Kristen Boyle[90], Tina Costacou[58], John M. Dennis[28], Jose C. Florez[20,21,32,33], Anna L. Gloyn[44,45,174], Maria F. Gomez[22,175], Peter A. Gottlieb[133], Siri Atma W. Greeley[176], Kurt Griffin[122,177], Andrew T. Hattersley[28,106], Irl B. Hirsch[178], Marie-France Hivert[20,179,180], Korey K. Hood[81], Jami L. Josefson[162], Soo Heon Kwak[181], Lori M. Laffel[182], Siew S. Lim[1], Ruth J. F. Loos[19,183], Ronald C. W. Ma[54,76,77], Chantal Mathieu[39], Nestoras Mathioudakis[65], James B. Meigs[33,108,184], Shivani Misra[185,186], Viswanathan Mohan[187], Rinki Murphy[188,189,190], Richard Oram[28,106], Katharine R. Owen[107,191], Susan E. Ozanne[192], Ewan R. Pearson[43], Wei Perng[90], Toni I. Pollin[152,193], Rodica Pop-Busui[194], Richard E. Pratley[195], Leanne M. Redman[3,204], Maria J. Redondo[78,79], Rebecca M. Reynolds[57], Robert K. Semple[57,196], Jennifer L. Sherr[197], Emily K. Sims[35,36,37], Arianne Sweeting[198,199], Tiinamaija Tuomi[48,145,50], Miriam S. Udler[20,21,32,33], Kimberly K. Vesco[200], Tina Vilsbøll[201,202], Robert Wagner[67,68,203], Stephen S. Rich[116] & Paul W. Franks[18,22,107,114]

[17]Division of Preventative Medicine, Department of Medicine, Brigham and Women's Hospital and Harvard Medical School, Boston, MA, USA. [18]Department of Nutrition, Harvard T.H. Chan School of Public Health, Boston, MA, USA. [19]Novo Nordisk Foundation Center for Basic Metabolic Research, Faculty of Health and Medical Sciences, University of Copenhagen, Copenhagen, Denmark. [20]Diabetes Unit, Endocrine Division, Massachusetts General Hospital, Boston, MA, USA. [21]Center for Genomic Medicine, Massachusetts General Hospital, Boston, MA, USA. [22]Department of Clinical Sciences, Lund University Diabetes Centre, Lund University Malmö, Sweden. [23]Department of Obstetrics and Gynaecology, the Rosie Hospital, Cambridge, UK. [24]NIHR Cambridge Biomedical Research Centre, University of Cambridge, Cambridge, UK. [25]Departments of Medicine and Community Health Sciences, Cumming School of Medicine, University of Calgary, Calgary, AB, Canada. [26]Department of Molecular Genetics, Madras Diabetes Research Foundation, Chennai, India. [27]Division of Pediatric Endocrinology, Department of Pediatrics, Saint Louis University School of Medicine, SSM Health Cardinal Glennon Children's Hospital, St. Louis, MO, USA. [28]Department of Clinical and Biomedical Sciences, University of Exeter Medical School, Exeter Devon, UK. [29]CIBER-BBN, ISCIII, Madrid, Spain. [30]Institut d'Investigació Biomèdica Sant Pau (IIB SANT PAU), Barcelona, Spain. [31]Departament de Medicina, Universitat Autònoma de Barcelona, Bellaterra, Spain. [32]Programs in Metabolism and Medical & Population Genetics, Broad Institute, Cambridge, MA, USA. [33]Department of Medicine, Harvard Medical School, Boston, MA, USA. [34]Division of Endocrinology, Diabetes and Metabolism, Johns Hopkins University School of Medicine, Baltimore, MD, USA. [35]Department of Pediatrics, Indiana University School of Medicine, Indianapolis, IN, USA. [36]Herman B Wells Center for Pediatric Research, Indiana University School of Medicine, Indianapolis, IN, USA. [37]Center for Diabetes and Metabolic Diseases, Indiana University School of Medicine, Indianapolis, IN, USA. [38]Department of Biostatistics and Epidemiology, Rutgers School of Public Health, Piscataway, NJ, USA. [39]University Hospital Leuven, Leuven, Belgium. [40]Department of Nutrition, Université de Montréal, Montreal, Quebec, Canada. [41]Research Center, Sainte-Justine University Hospital Center, Montreal, Quebec, Canada. [42]Department of Pediatrics, Erasmus Medical Center, Rotterdam, The Netherlands. [43]Division of Population Health & Genomics, School of Medicine, University of Dundee, Dundee, UK. [44]Department of Pediatrics, Stanford School of Medicine, Stanford University, Stanford, CA, USA. [45]Stanford Diabetes Research Center, Stanford School of Medicine, Stanford University, Stanford, CA, USA. [46]University of Florida, Gainesville, FL, USA. [47]Department of Nutrition, University of North Carolina at Chapel Hill, Chapel Hill, NC, USA. [48]Helsinki University Hospital, Abdominal Centre/Endocrinology, Helsinki, Finland. [49]Folkhalsan Research Center, Helsinki, Finland. [50]Institute for Molecular Medicine Finland FIMM, University of Helsinki, Helsinki, Finland. [51]Department of Pediatrics, Division of

Endocrinology, Boston Children's Hospital, Boston, MA, USA. [52]Department of Medicine, Faculty of Medicine, University of Malaya, Kuala Lumpur, Malaysia. [53]Asia Diabetes Foundation, Hong Kong SAR, China. [54]Department of Medicine & Therapeutics, Chinese University of Hong Kong, Hong Kong SAR, China. [55]Departments of Pediatrics and Clinical Genetics, Kuopio University Hospital, Kuopio, Finland. [56]Department of Medicine, University of Eastern Finland, Kuopio, Finland. [57]Centre for Cardiovascular Science, Queen's Medical Research Institute, University of Edinburgh, Edinburgh, UK. [58]Department of Epidemiology, University of Pittsburgh, Pittsburgh, PA, USA. [59]Metabolic Disease Unit, University Hospital of Padova, Padova, Italy. [60]Department of Medicine, University of Padova, Padova, Italy. [61]Department of Orthopedics, Zuyderland Medical Center, Sittard-Geleen, The Netherlands. [62]Departments of Pediatrics and Medicine, University of Chicago, Chicago, Illinois, USA. [63]Welch Center for Prevention, Epidemiology, and Clinical Research, Johns Hopkins Bloomberg School of Public Health, Baltimore, Maryland, USA. [64]Ciccarone Center for the Prevention of Cardiovascular Disease, Johns Hopkins School of Medicine, Baltimore, MD, USA. [65]Department of Medicine, Johns Hopkins University, Baltimore, MD, USA. [66]Department of Health Policy and Management, Johns Hopkins University Bloomberg School of Public Health, Baltimore, Maryland, USA. [67]Institute for Clinical Diabetology, German Diabetes Center, Leibniz Center for Diabetes Research at Heinrich Heine University Düsseldorf, Auf'm Hennekamp 65, 40225 Düsseldorf, Germany. [68]German Center for Diabetes Research (DZD), Ingolstädter Landstraße 1, 85764 Neuherberg, Germany. [69]Section of Academic Primary Care, US Department of Veterans Affairs Eastern Colorado Health Care System, Aurora, CO, USA. [70]Department of Medicine, University of Colorado School of Medicine, Aurora, CO, USA. [71]Department of Epidemiology, Johns Hopkins Bloomberg School of Public Health, Baltimore, Maryland, USA. [72]Institute of Experimental Endocrinology, Biomedical Research Center, Slovak Academy of Sciences, Bratislava, Slovakia. [73]Clinical and Translational Epidemiology Unit, Massachusetts General Hospital, Boston, MA, USA. [74]Mohn Center for Diabetes Precision Medicine, Department of Clinical Science, University of Bergen, Bergen, Norway. [75]Children and Youth Clinic, Haukeland University Hospital, Bergen, Norway. [76]Laboratory for Molecular Epidemiology in Diabetes, Li Ka Shing Institute of Health Sciences, The Chinese University of Hong Kong, Hong Kong, China. [77]Hong Kong Institute of Diabetes and Obesity, The Chinese University of Hong Kong, Hong Kong, China. [78]Department of Pediatrics, Baylor College of Medicine, Houston, TX, USA. [79]Division of Pediatric Diabetes and Endocrinology, Texas Children's Hospital, Houston, TX, USA. [80]Children's Nutrition Research Center, USDA/ARS, Houston, TX, USA. [81]Stanford University School of Medicine, Stanford, CA, USA. [82]Internal Medicine, University of Manitoba, Winnipeg, MB, Canada. [83]Department of Diabetology, APHP, Paris, France. [84]Sorbonne Université, INSERM, NutriOmic team, Paris, France. [85]Department of Nutrition, Dietetics and Food, Monash University, Melbourne, Victoria, Australia. [86]Health Management Center, The Second Affiliated Hospital of Chongqing Medical University, Chongqing Medical University, Chongqing, China. [87]MRC/Wits Developmental Pathways for Health Research Unit, Department of Paediatrics, Faculty of Health Sciences, University of the Witwatersrand, Johannesburg, South Africa. [88]Channing Division of Network Medicine, Brigham and Women's Hospital, Boston, MA, USA. [89]Sydney Brenner Institute for Molecular Bioscience, Faculty of Health Sciences, University of the Witwatersrand, Johannesburg, South Africa. [90]Lifecourse Epidemiology of Adiposity and Diabetes (LEAD) Center, University of Colorado Anschutz Medical Campus, Aurora, CO, USA. [91]Section of Adult and Pediatric Endocrinology, Diabetes and Metabolism, Kovler Diabetes Center, University of Chicago, Chicago, USA. [92]Department of Pediatrics, Riley Hospital for Children, Indiana University School of Medicine, Indianapolis, IN, USA. [93]Richard L. Roudebush VAMC, Indianapolis, IN, USA. [94]Biomedical Research Institute Girona, IdIBGi, Girona, Spain. [95]Diabetes, Endocrinology and Nutrition Unit Girona, University Hospital Dr Josep Trueta, Girona, Spain. [96]Institute of Health System Science, Feinstein Institutes for Medical Research, Northwell Health, Manhasset, NY, USA. [97]University of California at San Francisco, Department of Pediatrics, Diabetes Center, San Francisco, CA, USA. [98]Division of Endocrinology, Diabetes and Metabolism, Cedars-Sinai Medical Center, Los Angeles, CA, USA. [99]Department of Medicine, Cedars-Sinai Medical Center, Los Angeles, CA, USA. [100]Robinson Research Institute, The University of Adelaide, Adelaide, Australia. [101]Department of Public Health and Novo Nordisk Foundation Center for Basic Metabolic Research, Faculty of Health and Medical Sciences, University of Copenhagen, 1014 Copenhagen, Denmark. [102]Division of Endocrinology and Diabetes, Department of Pediatrics, Sanford Children's Hospital, Sioux Falls, SD, USA. [103]University of South Dakota School of Medicine, E Clark St, Vermillion, SD, USA. [104]Department of Biomedical Informatics, University of Colorado Anschutz Medical Campus, Aurora, CO, USA. [105]Department of Epidemiology, Colorado School of Public Health, Aurora, CO, USA. [106]Royal Devon University Healthcare NHS Foundation Trust, Exeter, UK. [107]Oxford Centre for Diabetes, Endocrinology and Metabolism, University of Oxford, Oxford, UK. [108]Division of General Internal Medicine, Massachusetts General Hospital, Boston, MA, USA. [109]UPMC Children's Hospital of Pittsburgh, Pittsburgh, PA, USA. [110]Center for Translational Immunology, Benaroya Research Institute, Seattle, WA, USA. [111]Department of Medicine, Northwestern University Feinberg School of Medicine, Chicago, IL, USA. [112]Department of Pathology & Molecular Medicine, McMaster University, Hamilton, Canada. [113]Population Health Research Institute, Hamilton, Canada. [114]Department of Translational Medicine, Medical Science, Novo Nordisk Foundation, Tuborg Havnevej 19, 2900 Hellerup, Denmark. [115]Department of Diabetes and Endocrinology, Nelson R Mandela School of Medicine, University of KwaZulu-Natal, Durban, South Africa. [116]Center for Public Health Genomics, Department of Public Health Sciences, University of Virginia, Charlottesville, VA, USA. [117]Division of Epidemiology and Community Health, School of Public Health, University of Minnesota, Minneapolis, MN, USA. [118]Department of Chronic Diseases and Metabolism, Clinical and Experimental Endocrinology, KU Leuven, Leuven, Belgium. [119]School of Health and Wellbeing, College of Medical, Veterinary and Life Sciences, University of Glasgow, Glasgow, UK. [120]Department of Obstetrics, Gynecology, and Reproductive Biology, Massachusetts General Hospital and Harvard Medical School, Boston, MA, USA. [121]Sanford Children's Specialty Clinic, Sioux Falls, SD, USA. [122]Department of Pediatrics, Sanford School of Medicine, University of South Dakota, Sioux Falls, SD, USA. [123]Department of Biostatistics, Johns Hopkins Bloomberg School of Public Health, Baltimore, Maryland, USA. [124]Centre for Physical Activity Research, Rigshospitalet, Copenhagen, Denmark. [125]Institute for Sports and Clinical Biomechanics, University of Southern Denmark, Odense, Denmark. [126]Department of Medicine, Division of Endocrinology, Diabetes and Metabolism, Indiana University School of Medicine, Indianapolis, IN, USA. [127]AMAN Hospital, Doha, Qatar. [128]Department of Preventive Medicine, Division of Biostatistics, Northwestern University Feinberg School of Medicine, Chicago, IL, USA. [129]Institute of Molecular and Genomic Medicine, National Health Research Institutes, Taipei City, Taiwan. [130]Divsion of Endocrinology and Metabolism, Taichung Veterans General Hospital, Taichung, Taiwan. [131]Division of Endocrinology and Metabolism, Taipei Veterans General Hospital, Taipei, Taiwan. [132]Center for Interventional Immunology, Benaroya Research Institute, Seattle, WA, USA. [133]Barbara Davis Center for Diabetes, University of Colorado Anschutz Medical Campus, Aurora, CO, USA. [134]University Hospital of Tübingen, Tübingen, Germany. [135]Institute of Diabetes Research and Metabolic Diseases (IDM), Helmholtz Center Munich, Neuherberg, Germany. [136]Steno Diabetes Center Aarhus, Aarhus University Hospital, Aarhus, Denmark. [137]University of Newcastle, Newcastle upon Tyne, UK. [138]Sections on Genetics and Epidemiology, Joslin Diabetes Center, Harvard Medical School, Boston, MA, USA. [139]Department of Clinical Pharmacy and Pharmacology, University Medical Center Groningen, Groningen, The Netherlands. [140]Gastroenterology, Baylor College of Medicine, Houston, TX, USA. [141]Department of Endocrinology, University Hospitals Leuven, Leuven, Belgium. [142]Sorbonne University, Inserm U938, Saint-Antoine Research Centre, Institute of Cardiometabolism and Nutrition, Paris 75012, France. [143]Department of Endocrinology, Diabetology and Reproductive Endocrinology, Assistance Publique-Hôpitaux de Paris, Saint-Antoine University Hospital, National Reference Center for Rare Diseases of Insulin Secretion and Insulin Sensitivity (PRISIS), Paris, France. [144]Royal Melbourne Hospital Department of Diabetes and Endocrinology, Parkville, Vic, Australia. [145]Walter and Eliza Hall Institute, Parkville, Vic, Australia. [146]University of Melbourne Department of Medicine, Parkville, Vic, Australia. [147]Department of Epidemiology, Madras Diabetes Research Foundation, Chennai, India.

[148]Department of Diabetes and Endocrinology, Guy's and St Thomas' Hospitals NHS Foundation Trust, London, UK. [149]Institut Cochin, Inserm U 10116 Paris, France. [150]Pediatric endocrinology and diabetes, Hopital Necker Enfants Malades, APHP Centre, université de Paris, Paris, France. [151]Department of Medical Genetics, Haukeland University Hospital, Bergen, Norway. [152]Department of Medicine, University of Maryland School of Medicine, Baltimore, MD, USA. [153]Department of Epidemiology, Geisel School of Medicine at Dartmouth, Hanover, NH, USA. [154]Nephrology, Dialysis and Renal Transplant Unit, IRCCS—Azienda Ospedaliero-Universitaria di Bologna, Alma Mater Studiorum University of Bologna, Bologna, Italy. [155]Department of Medical Genetics, AP-HP Pitié-Salpêtrière Hospital, Sorbonne University, Paris, France. [156]Global Center for Asian Women's Health, Yong Loo Lin School of Medicine, National University of Singapore, Singapore, Singapore. [157]Department of Obstetrics and Gynecology, Yong Loo Lin School of Medicine, National University of Singapore, Singapore, Singapore. [158]Kaiser Permanente Northern California Division of Research, Oakland, California, USA. [159]Department of Epidemiology and Biostatistics, University of California San Francisco, California, USA. [160]National Institute of Diabetes and Digestive and Kidney Diseases, National Institutes of Health, Bethesda, MD, USA. [161]Department of Health Research Methods, Evidence, and Impact, Faculty of Health Sciences, McMaster University, Hamilton, ON, Canada. [162]Ann & Robert H. Lurie Children's Hospital of Chicago, Department of Pediatrics, Northwestern University Feinberg School of Medicine, Chicago, IL, USA. [163]Department of Clinical and Organizational Development, Chicago, IL, USA. [164]American Diabetes Association, Arlington, Virginia, USA. [165]College of Medicine and Health Sciences, University of Gondar, Gondar, Ethiopia. [166]Global Health Institute, Faculty of Medicine and Health Sciences, University of Antwerp, 2160 Antwerp, Belgium. [167]Department of Medicine and Kovler Diabetes Center, University of Chicago, Chicago, IL, USA. [168]School of Nursing, Faculty of Health Sciences, McMaster University, Hamilton, Canada. [169]Division of Endocrinology, Metabolism, Diabetes, University of Colorado, Boulder, CO, USA. [170]Department of Clinical Medicine, School of Medicine, Trinity College Dublin, Dublin, Ireland. [171]Department of Endocrinology, Wexford General Hospital, Wexford, Ireland. [172]Division of Endocrinology, NorthShore University HealthSystem, Skokie, IL, USA. [173]Department of Medicine, Prtizker School of Medicine, University of Chicago, Chicago, IL, USA. [174]Department of Genetics, Stanford School of Medicine, Stanford University, Stanford, CA, USA. [175]Faculty of Health, Aarhus University, Aarhus, Denmark. [176]Departments of Pediatrics and Medicine and Kovler Diabetes Center, University of Chicago, Chicago, USA. [177]Sanford Research, Sioux Falls, SD, USA. [178]University of Washington School of Medicine, Seattle, WA, USA. [179]Department of Population Medicine, Harvard Medical School, Harvard Pilgrim Health Care Institute, Boston, MA, USA. [180]Department of Medicine, Universite de Sherbrooke, Sherbrooke, QC, Canada. [181]Department of Internal Medicine, Seoul National University College of Medicine, Seoul National University Hospital, Seoul, Republic of Korea. [182]Joslin Diabetes Center, Harvard Medical School, Boston, MA, USA. [183]Charles Bronfman Institute for Personalized Medicine, Icahn School of Medicine at Mount Sinai, New York, NY, USA. [184]Broad Institute, Cambridge, MA, USA. [185]Division of Metabolism, Digestion and Reproduction, Imperial College London, London, UK. [186]Department of Diabetes & Endocrinology, Imperial College Healthcare NHS Trust, London, UK. [187]Department of Diabetology, Madras Diabetes Research Foundation & Dr. Mohan's Diabetes Specialities Centre, Chennai, India. [188]Department of Medicine, Faculty of Medicine and Health Sciences, University of Auckland, Auckland, New Zealand. [189]Auckland Diabetes Centre, Te Whatu Ora Health New Zealand, Auckland, New Zealand. [190]Medical Bariatric Service, Te Whatu Ora Counties, Health New Zealand, Auckland, New Zealand. [191]Oxford NIHR Biomedical Research Centre, University of Oxford, Oxford, UK. [192]University of Cambridge, Metabolic Research Laboratories and MRC Metabolic Diseases Unit, Wellcome-MRC Institute of Metabolic Science, Cambridge, UK. [193]Department of Epidemiology & Public Health, University of Maryland School of Medicine, Baltimore, MD, USA. [194]Department of Internal Medicine, Division of Metabolism, Endocrinology and Diabetes, University of Michigan, Ann Arbor, MI, USA. [195]AdventHealth Translational Research Institute, Orlando, FL, USA. [196]MRC Human Genetics Unit, Institute of Genetics and Cancer, University of Edinburgh, Edinburgh, UK. [197]Yale School of Medicine, New Haven, CT, USA. [198]Faculty of Medicine and Health, University of Sydney, Sydney, NSW, Australia. [199]Department of Endocrinology, Royal Prince Alfred Hospital, Sydney, NSW, Australia. [200]Kaiser Permanente Northwest, Kaiser Permanente Center for Health Research, Portland, OR, USA. [201]Clinial Research, Steno Diabetes Center Copenhagen, Herlev, Denmark. [202]Department of Clinical Medicine, Faculty of Health and Medical Sciences, University of Copenhagen, Copenhagen, Denmark. [203]Department of Endocrinology and Diabetology, University Hospital Düsseldorf, Heinrich Heine University Düsseldorf, Moorenstr. 5, 40225 Düsseldorf, Germany.

