## [Peer Review File · Communications Medicine]

Reviewers' comments:

Reviewer #1 (Remarks to the Author):

The notion of using a range of characteristics – genetic/clinical/ psychosocial/biochemical as markers to determine responses to preventative strategies for GDM and T2DM is very appealing. For this reason this systematic review is timely and constitutes a huge body of work, but it merely serves to demonstrate the weakness of many of the studies and the lack of inclusion of a standard list of variables in the studies that could enable the development of determinants for trials for GDM prevention in a meaningful way.

It is unclear to me why studies with a sample size of < 100 are included- would it not be better to insist on a minimal sample size?

Would the authors provide some explanation as to why physical activity had a larger effect in non obese than obese women

The conclusions should be tempered to reflect on the major deficiencies in the outcomes of the review and where it takes the drive for precision medicine for GDM prevention.

Reviewer #2 (Remarks to the Author):

I like this paper (always good to start there).

The underlying concept of application of precision medicine to this area of research and clinical medicine is highly relevant and very topical.

The paper contains substantial information that would be of enormous value to future researchers.

The systematic review 'process' has been well followed according to the appropriate 'recipe' for such work and I have no criticisms regarding methodology.

I think that there are two 'layers' to this paper, and if I have a criticism it is that excessive adherence to the former reduces the impact of the latter:

1. The paper has substantial information that is derived in a highly appropriate manner and is well presented. However this is quite 'dense' and makes the paper read as more of an exercise in how to do an excellent systematic review. It does make the paper a bit impenetrable and I worry that this will reduce impact.
2. There is a more external layer to this paper which relates to the growing area of precision medicine, what this means in this area and how this is highly relevant for GDM. This is somewhat buried in all of the (highly relevant) detail that is included.

I would make a couple of particular points here:

1. It actually took me a little while to 'work out' exactly what the authors were aiming to present and why. In part that could be because the concept of precision medicine is relatively emerging albeit highly relevant. I think that a stronger focus on why this is a significant future direction for medicine could be of benefit. This would lead the reader in to understanding the 'framing' of this work and its relevance.
2. The final sentence of the Introduction really sums up what this work is all about and it would be good to make this more 'front and centre' for readers.
3. The Discussion is good, but also relatively impenetrable. I think that there could be more general focus on reminding a reviewer of *why* the knowledge gained from this work is of great value.

My apologies if my comments are more 'conceptual' than specific. It would be good to see this work published but a review by the senior authors to focus on the message and its presentation could make it a much more readable piece of work. I'd be very happy to review this again.

More specific individual comments are included in an attached document.

Reviewer #3 (Remarks to the Author):

The systematic review and meta-analysis of Lim et al. covers a relevant topic. The review has been conducted according to current standards and the results are reported clearly, in general. The findings add to our knowledge on the most effective strategies for GDM prevention. The paper is concise and well-written.

Major issues:

- I have some problems with the last sentence of the abstract and conclusion, that future research should provide results stratified by participant characteristics. I fully agree with the authors that relevant interactions for intervention effects should be assessed, but how it is formulated at the moment, it seems as if researchers are invited to go on a fishing expedition. In my opinion, it should be based on a-priori formulated plans for subgroup analyses. I am sure the authors have the same opinion, and can formulate their suggestion a bit differently.
- How was it decided whether there was a difference in effect between the subgroups? For an original study, one would assess whether the interaction between the variable of interest and intervention group is significant. How has this been done here? It seems there was some rationale behind it, since not all subgroups are presented for all types of intervention, and it is important to understand this decision.
- Why were not all intervention arms included in the meta-analysis of some trials (for instance, the DALI study reported in Simmons et al. or the fish oil study from Pellonpera et al)? Those other study arms are not mentioned in line 210, and not all seem to be included in the meta-analyses.
- Based on the information provided in Suppl Table 4, I get the impression that for some trials the original (effect) publication was not used, since many items (e.g. smoking, previous GDM etc) were reported in the original publications, but in the table, it says "NR". This is true, for instance, for Simmons et al. (DALI), Rönö et al. (RADIEL).
- In the discussion, it could be mentioned that (federated) individual level meta-analyses might be useful for identifying relevant subgroups who respond differently to interventions. This would not have the limitations of the current meta-analysis, as acknowledged by the authors.

Minor issues:

- Line 113: I think that Physical Activity is more appropriate in this context than exercise. Please replace
- Line 133: is postpartum really relevant for this review? I did not see any studies in the postpartum period included in the review.
- Table 1: Why is the RADIEL study listed under non-randomised controlled trials? As far as I know, this study had random allocation to intervention and control group.
- Supplementary materials need to be rechecked for spelling and correctness (for instance heading of Suppl Figure 1: GWG should be GDM)

Since the differences between information provided in this manuscript and the original publications were only identified for the trials that I am most familiar with, this does not preclude similar issues for the other trials included in this systematic review. Please carefully check all information again.

Reviewer #1 (Remarks to the Author):

The notion of using a range of characteristics – genetic/clinical/ psychosocial/biochemical as markers to determine responses to preventative strategies for GDM and T2DM is very appealing. For this reason this systematic review is timely and constitutes a huge body of work, but it merely serves to demonstrate the weakness of many of the studies and the lack of inclusion of a standard list of variables in the studies that could enable the development of determinants for trials for GDM prevention in a meaningful way.

Thank you for the comment.

It is unclear to me why studies with a sample size of < 100 are included- would it not be better to insist on a minimal sample size?

According to the Cochrane handbook, one of the purposes of meta-analysis is to improve precision for studies that are too small to provide conclusive evidence on intervention effects in isolation (Higgins & Thomas, 2020). It is therefore not a convention for meta-analyses to impose a limit by sample size.

Would the authors provide some explanation as to why physical activity had a larger effect in non obese than obese women

The following has been added to the paragraph in Discussion explaining lifestyle interventions on GDM prevention:

Further, we found that physical activity interventions were only effective in individuals with normal BMI but not in those with obese BMI. Similar observations were previously reported in other meta-analyses of GDM prevention in individuals with excess body weight (Muhammad et al., 2021; Nasiri-Amiri et al., 2019). As obesity is associated with increased insulin resistance, it is possible that physical activity interventions alone may not be sufficient to lower GDM risks in these individuals.

The conclusions should be tempered to reflect on the major deficiencies in the outcomes of the review and where it takes the drive for precision medicine for GDM prevention.

The following were added (underlined) to better reflect the limitations in the concluding paragraph.:

However, these results should be interpreted with caution due to limited reporting of intervention outcomes by participant characteristics in the individual studies. Given the potentially greater effectiveness of lifestyle and metformin interventions in certain individuals, future research on tailored recommendations in precision GDM prevention, replacing the current 'one-size-fits-all' approach, is needed.

Reviewer #2 (Remarks to the Author):

I like this paper (always good to start there).

The underlying concept of application of precision medicine to this area of research and clinical medicine is highly relevant and very topical.

The paper contains substantial information that would be of enormous value to future researchers. The systematic review 'process' has been well followed according to the appropriate 'recipe' for such work and I have no criticisms regarding methodology.

Thank you for the comment.

I think that there are two 'layers' to this paper, and if I have a criticism it is that excessive adherence to the former reduces the impact of the latter:

1. The paper has substantial information that is derived in a highly appropriate manner and is well presented. However this is quite 'dense' and makes the paper read as more of an exercise in how to do an excellent systematic review. It does make the paper a bit impenetrable and I worry that this will reduce impact.
2. There is a more external layer to this paper which relates to the growing area of precision medicine, what this means in this area and how this is highly relevant for GDM. This is somewhat buried in all of the (highly relevant) detail that is included.

Thank you for sharing this helpful insight as a reader. We agree this paper provides a large amount of data as we reviewed > 100 studies. We made a great effort in the Conclusion section to summarize the important findings from this systematic review.

To address point 2: The last 2 sentences of the conclusion place this paper in context of precision medicine objectives and summarize the goals of precision medicine among individuals at risk of GDM.

I hope by addressing the specific points below we have made some improvements in this regard.

I would make a couple of particular points here:

1. It actually took me a little while to 'work out' exactly what the authors were aiming to present and why. In part that could be because the concept of precision medicine is relatively emerging albeit highly relevant. I think that a stronger focus on why this is a significant future direction for medicine could be of benefit. This would lead the reader in to understanding the 'framing' of this work and its relevance.

Please see below the response for Point #2.

2. The final sentence of the Introduction really sums up what this work is all about and it would be good to make this more 'front and centre' for readers.

The following has been added in the introduction to strengthen the description of our framing and rationale for this work, placing the message of the final sentence of the Introduction front and centre for the readers:

Individual characteristics such as clinical, psychosocial and biochemical factors may influence the effectiveness of interventions in preventing GDM. The prevention of GDM results from an interaction between behavioural factors (such as the ability to adhere to the intervention) and physiological factors (such as the biological responsiveness towards reducing insulin resistance). Hence, clinical traits including overweight/obesity, age, history of GDM or polycystic ovary syndrome (PCOS) and others, along with social determinants of health including socioeconomic status, cultural background,

race or ethnicity and others, are potential sources of heterogeneity in intervention effect. These clinical, biochemical, social and environmental traits could affect GDM prevention through behavioural or physiological pathways, or both. Given that interventions to prevent GDM are unlikely to be effective for every individual alike, there is a need to elucidate the most effective mode of prevention for each population. To date, there has not been a comprehensive meta-analysis of GDM prevention, accounting for participant characteristics to inform precision medicine.

3. The Discussion is good, but also relatively impenetrable. I think that there could be more general focus on reminding a reviewer of *why* the knowledge gained from this work is of great value.

The following changes were made to increase the readability of Discussion, and to centre the key messages for the readers:

Further research is needed to determine if more intensive lifestyle intervention or additional co-intervention such as metformin or supplementation is needed to prevent GDM in women with conditions of high insulin resistance including prior GDM, PCOS and/or obese body weight. This is important so as not to provide unfounded expectations on the benefit of lifestyle on GDM prevention for certain groups. It is also of particular pertinence to these specific populations outlined, given the stigma associated with obesity and diabetes, resulting from the perception that these health outcomes are caused by personal failures (Hill & Incollingo Rodriguez, 2020; Speight & Holmes-Truscott, 2023).

Our findings suggest the metformin may be the intervention of choice for preventing GDM in populations at high risk of insulin resistance, including in women with advanced age, higher fasting blood glucose, history of GDM and/or PCOS, along with healthy antenatal lifestyle advice. However, we found that the quality of evidence for metformin in preventing GDM was very low, thus further high-quality RCTs are needed to confirm these findings.

In this review, we identified only seven preconception interventions that met our inclusion criteria. Despite this limitation, we found that lifestyle or metformin interventions were more effective in lowering the risk for GDM when commenced preconception or in the first trimester of pregnancy.

My apologies if my comments are more 'conceptual' than specific. It would be good to see this work published but a review by the senior authors to focus on the message and its presentation could make it a much more readable piece of work. I'd be very happy to review this again.

Thank you—your comments are very helpful and insightful and we hope our revised manuscript will increase the clarity of the message of this paper.

More specific individual comments are included in an attached document.

1.

Line 101 - "The physiologic reduction of insulin sensitivity during pregnancy is the hallmark metabolic feature that leads to the onset of glucose intolerance and GDM in predisposed subjects."

Changed as suggested. In the manuscript I changed subjects to individuals

2.

Line 102-103: ~~older advanced~~ maternal age, parity, ~~being~~ overweight/obese ~~body mass index (BMI)~~, and family history of diabetes.

This has been changed to below, with consideration of people-first language:

Established risk factors for GDM include previous GDM, advanced maternal age, parity, overweight/obesity, and family history of diabetes (Habibi et al., 2022; Zhang et al., 2021)

3.

Line 131-135: This Systematic Review and subsequent meta-analysis examined the effectiveness of interventions employing lifestyle modification, metformin, or dietary supplements within the preconception, pregnant and postpartum periods for reducing the risk of developing GDM. ~~The review is written on behalf of the ADA/EASD PMDI as part of a comprehensive evidence evaluation in support of the 2nd International Consensus Report on Precision Diabetes Medicine [crossref Tobias et al, Nat Med].~~

Changed as suggested

4.

Line 150: We searched the following databases: Embase (Elsevier), Ovid Medline, and PubMed from ~~inception~~ ~~January~~ to May 24, 2022. Results were limited to ~~human~~ studies ~~based on human and in~~ English-language. ~~(NOTE NO TIMEFRAME OF PUBLICATIONS SPECIFIED)~~

This has been changed to the following:

We searched the following databases: Embase (Elsevier), Ovid Medline, and PubMed from the inception of the database to May 24, 2022. Results were limited to studies in human and in English-language. No limit was placed on publication date.

5.

Line 157-161: Randomised and non-randomized controlled trials (RCTs and non-RCTs) investigating the effects of lifestyle (diet, exercise, or both), metformin, or dietary supplements (fish oil, myoinositol/inositol, probiotics) on prevention of GDM in women of childbearing age (including preconception cohorts) were included (Supplementary Table 2). ~~Interventions included lifestyle (diet, exercise, or a combination of these interventions), metformin and dietary supplements.~~

Deleted as suggested

6.

Line 239:-Of the 54 studies which reported race, 18 studies were predominantly conducted among White participants, 10 were predominantly conducted among non-~~White~~ ~~caucasian~~ participants, and 26 among mixed populations.

Line 304 – 305: Subgroup analyses showed that myoinositol/303 inositol interventions were more effective in reducing GDM in studies involving mostly ~~White~~ ~~caucasian~~ women compared with women from various ethnic backgrounds

As the term Caucasian refers to people from the Caucasus region in Eurasia, it is recommended that we should use the term White instead (Flanagin et al., 2021).

7.

Line 242: The criteria used for GDM diagnosis varied broadly across the studies and included both one-step (a single oral glucose tolerance test (OGTT), most commonly a single 75-gram, 2-hour oral glucose tolerance test - OGTT) and two-step methods (commonly the 50 gram one-hour oral glucose challenge test, followed by a 2 or 3 hours OGTT if the oral glucose challenge test was abnormal).

This has been changed to the following:

The criteria used for GDM diagnosis varied broadly across the studies and included both one-step (most commonly a single 75-gram, 2-hour oral glucose tolerance test) and two-step methods (commonly the 50 gram one-hour oral glucose challenge test, followed by a 2 or 3-hour oral glucose tolerance test if the oral glucose challenge test was abnormal).

8.

Line 283: Diet and physical activity interventions were also more effective in reducing GDM in studies involving women without PCOS compared with to those with PCOS, and in studies involving women without a history of GDM compared with to those with unspecified history of GDM (Table 6).

Changed as suggested

9.

Line 356- 358: In addition, lower adherence to a healthy diet, as lower dietary quality, ~~has been observed~~ among women with a history of GDM compared with women without a history of GDM, may have also contributed to this (32).

Changed as suggested

10.

Line 372-374: By increasing the number of included metformin trials ~~studies~~ from 3 in the previous meta-analysis (16) to the current meta-analysis of 13 metformin trials, the current review revealed a significant reduction in GDM with metformin.

Changed as suggested

Reviewer #3 (Remarks to the Author):

The systematic review and meta-analysis of Lim et al. covers a relevant topic. The review has been conducted according to current standards and the results are reported clearly, in general. The findings add to our knowledge on the most effective strategies for GDM prevention. The paper is concise and well-written.

Thank you for the comment.

Major issues:

- I have some problems with the last sentence of the abstract and conclusion, that future research should provide results stratified by participant characteristics. I fully agree with the authors that relevant interactions for intervention effects should be assessed, but how it is formulated at the moment, it seems as if researchers are invited to go on a fishing expedition. In my opinion, it should

be based on a-priori formulated plans for subgroup analyses. I am sure the authors have the same opinion, and can formulate their suggestion a bit differently.

Disaggregation of participant groupings is key to understanding the effect of interventions on specific population groups, as highlighted in a recent Lancet series (Hassan et al., 2023). A stratified analysis is one of the many approaches to disaggregating research populations. To this end, we have changed “stratified” to “disaggregate” to better convey this message.

The Abstract was amended as below:

GDM prevention through metformin or lifestyle differs according to some individual characteristics. Future research should include trials commencing preconception and provide results disaggregated by participant characteristics including social and environmental factors, clinical traits, and other novel risk factors to predict GDM prevention through interventions.

The Conclusion was amended as below:

To advance knowledge in precision prevention, future research should include trials commencing in the preconception period and provide results disaggregated by participant characteristics, including social and environmental factors, clinical traits, and other novel risk factors.

- How was it decided whether there was a difference in effect between the subgroups? For an original study, one would assess whether the interaction between the variable of interest and intervention group is significant. How has this been done here? It seems there was some rationale behind it, since not all subgroups are presented for all types of intervention, and it is important to understand this decision.

Differences between the subgroups were indicated by the P-value between groups, as shown in one of columns in the Result Tables. It was determined using random-effects model as described in the Statistical analysis section. This site provides a user-friendly explanation of this methodology https://bookdown.org/MathiasHarrer/Doing_Meta_Analysis_in_R/subgroup.html. References were also provided in the manuscript for further exploration by readers. P-value between subgroups of less than 0.05 were interpreted as a significant difference between subgroups. Subgroup analyses were conducted whenever data was available for comparison. The following has been added to clarify these points in the Statistical Analysis section.

Subgroup analyses were conducted if there was at least one trial present in at least two comparative subgroups.

P<0.05 was taken as the level of statistical significance.

- Why were not all intervention arms included in the meta-analysis of some trials (for instance, the DALI study reported in Simmons et al. or the fish oil study from Pellonpera et al)? Those other study arms are not mentioned in line 210, and not all seem to be included in the meta-analyses.

When the protocol was developed, a decision was made to select only the most complex intervention arm for meta-analysis, in the case where multiple intervention arms are present in the same study that are equally relevant to the research question. This is to avoid duplicating the control group participants in the meta-analysis (Higgins & Thomas, 2020). However, in response to your comment,

we revisited this decision. As currently the meta-analyses were separately run for diet, exercise, and diet-and-exercise combined interventions, it would not introduce statistical error to include each intervention arm from these studies into the respective meta-analyses. Thus, we have now included additional diet-only and physical activity-only intervention arms from Renault et al (2014) and Simmons et al (2017) (Tables 1-5, p10, line 1-10).

For probiotic and fish-oil studies, there were insufficient number of studies to run meta-analysis for the intervention subtypes (e.g. probiotic-only, probiotic plus diet etc), hence these findings were provided as narrative synthesis in the Results section (p11, line 8-12).

- Based on the information provided in Suppl Table 4, I get the impression that for some trials the original (effect) publication was not used, since many items (e.g. smoking, previous GDM etc) were reported in the original publications, but in the table, it says "NR". This is true, for instance, for Simmons et al. (DALI), Rönö et al. (RADIEL).

The original intervention effects in these trials were captured in the **overall** meta-analysis for intervention effect. However, relatively few contributed to the **subgroup** analyses. To enable subgroup meta-analysis by participant characteristics for the intervention effects on GDM incidence, the individual study needed to provide **GDM incidence for each participant subgroup**, for the intervention and control arm. E.g. to perform a subgroup analysis by smoking status, we would need the number of GDM among smokers in the intervention group, the number of GDM among smokers in the control group, the number of GDM among non smokers in the intervention group, the number of GDM among non smokers in the control group.

Supplementary Table 4 on participant characteristics was intended to provide information to the readers on the categorisation of the participant characteristics that correspond to the subgroup analyses. This was provided to help with the interpretation of the subgroup analysis. NR was stated when there was no corresponding data provided on that participant characteristic for the subgroup analysis. We have now changed the title of the table to the following:

Supplementary Table 4: Categorisation of participant characteristics for subgroup analysis

- In the discussion, it could be mentioned that (federated) individual level meta-analyses might be useful for identifying relevant subgroups who respond differently to interventions. This would not have the limitations of the current meta-analysis, as acknowledged by the authors.

The following were added in the Strengths and Limitations section:

Participant characteristics that are known risk factors for GDM, such as parity, seldom feature in the inclusion or exclusion criteria of relevant studies, yielding a low number of subgroups available for comparison. This could be partly mitigated in individual level meta-analysis, if this information were collected at an individual level by the original studies.

Minor issues:

- Line 113: I think that Physical Activity is more appropriate in this context than exercise. Please replace

Thank you for bringing our attention to our inconsistency in the use of the terms Physical Activity and Exercise in the manuscript and tables. We have conducted a word check to ensure we use the term Physical Activity consistently throughout the manuscript and the tables. Physical activity include household and other incidental activities, while Exercise refers to planned activities to the aim of improving physical fitness (Caspersen et al., 1985). While most interventions in this review are likely Exercise interventions, it is difficult to preclude the increase of incidental physical activities due to the use of pedometers or lifestyle advice that may include increasing incidental physical activities. We have thus decided to use Physical Activity as a more inclusive term of all types of activities.

- Line 133: is postpartum really relevant for this review? I did not see any studies in the postpartum period included in the review.

When we developed the protocol, the authorship team of researchers and clinicians decided that postpartum studies provide important information on interconception/preconception interventions, which is highly relevant to the context of GDM prevention. Unfortunately, none of the postpartum studies met the inclusion criteria (they did not provide GDM incidence), as they tended to be type 2 diabetes prevention, instead of GDM prevention studies. To clarify our intent, this sentence has been modified to below:

This Systematic Review and subsequent meta-analysis examined the effectiveness of interventions employing lifestyle modification, metformin, or dietary supplements within the preconception, pregnant and postpartum/interconception periods for reducing the risk of developing GDM.

- Table 1: Why is the RADIEL study listed under non-randomised controlled trials? As far as I know, this study had random allocation to intervention and control group.

The Radiel study was misclassified due to the reporting of one of the secondary papers. We have now rechecked the classification of all the studies and have moved the Radiel study and a metformin study (Ainuddin 2015 et al) to the right categories.

- Supplementary materials need to be rechecked for spelling and correctness (for instance heading of Suppl Figure 1: GWG should be GDM)

The supplementary materials have now been checked for spelling and correctness. Formatting were standardised and spelling errors were corrected.

Since the differences between information provided in this manuscript and the original publications were only identified for the trials that I am most familiar with, this does not preclude similar issues for the other trials included in this systematic review. Please carefully check all information again.

We have carefully checked all the trials included in this systematic review for accuracy.

References:

- Caspersen, C. J., Powell, K. E., & Christenson, G. M. (1985, Mar-Apr). Physical activity, exercise, and physical fitness: definitions and distinctions for health-related research. *Public Health Rep, 100*(2), 126-131.
- Flanagin, A., Frey, T., & Christiansen, S. L. (2021, Aug 17). Updated Guidance on the Reporting of Race and Ethnicity in Medical and Science Journals. *JAMA, 326*(7), 621-627. <https://doi.org/10.1001/jama.2021.13304>
- Habibi, N., Mousa, A., Tay, C. T., Khomami, M. B., Patten, R. K., Andraweera, P. H., Wassie, M., Vandersluys, J., Aflatounian, A., Bianco-Miotto, T., Zhou, S. J., & Grieger, J. A. (2022, Jul). Maternal metabolic factors and the association with gestational diabetes: A systematic review and meta-analysis. *Diabetes Metab Res Rev, 38*(5), e3532. <https://doi.org/10.1002/dmrr.3532>
- Hassan, S., Gujral, U. P., Quarells, R. C., Rhodes, E. C., Shah, M. K., Obi, J., Lee, W. H., Shamambo, L., Weber, M. B., & Narayan, K. M. V. (2023, Jul). Disparities in diabetes prevalence and management by race and ethnicity in the USA: defining a path forward. *Lancet Diabetes Endocrinol, 11*(7), 509-524. [https://doi.org/10.1016/s2213-8587\(23\)00129-8](https://doi.org/10.1016/s2213-8587(23)00129-8)
- Higgins, J., & Thomas, J. (2020). *Cochrane Handbook for Systematic Reviews of Interventions, Version 6.1*. <https://training.cochrane.org/handbook/current>
- Hill, B., & Incollingo Rodriguez, A. C. (2020, Nov). Weight Stigma across the Preconception, Pregnancy, and Postpartum Periods: A Narrative Review and Conceptual Model. *Semin Reprod Med, 38*(6), 414-422. <https://doi.org/10.1055/s-0041-1723775>
- Muhammad, H. F. L., Pramono, A., & Rahman, M. N. (2021, Apr). The safety and efficacy of supervised exercise on pregnant women with overweight/obesity: A systematic review and meta-analysis of randomized controlled trials. *Clin Obes, 11*(2), e12428. <https://doi.org/10.1111/cob.12428>
- Nasiri-Amiri, F., Sepidarkish, M., Shirvani, M. A., Habibipour, P., & Tabari, N. S. M. (2019). The effect of exercise on the prevention of gestational diabetes in obese and overweight pregnant women: a systematic review and meta-analysis. *Diabetol Metab Syndr, 11*, 72. <https://doi.org/10.1186/s13098-019-0470-6>
- Speight, J., & Holmes-Truscott, E. (2023, Apr 17). Challenging diabetes stigma starts and ends with all of us. *Lancet Diabetes Endocrinol*. [https://doi.org/10.1016/s2213-8587\(23\)00084-0](https://doi.org/10.1016/s2213-8587(23)00084-0)
- Zhang, Y., Xiao, C. M., Zhang, Y., Chen, Q., Zhang, X. Q., Li, X. F., Shao, R. Y., & Gao, Y. M. (2021). Factors Associated with Gestational Diabetes Mellitus: A Meta-Analysis. *J Diabetes Res, 2021*, 6692695. <https://doi.org/10.1155/2021/6692695>

REVIEWERS' COMMENTS:

Reviewer #1 (Remarks to the Author):

I have no further comments

Reviewer #2 (Remarks to the Author):

I'm happy with the changes that have been made; in particular these changes make the overall 'message' easier to follow.

As a point of note, I'm not sure that the response to reviewers should necessarily be labelled a 'Rebuttal'. That expression suggests a structured argument against the reviewers, when usually the process would be more that of working with the reviewers.

Reviewer #3 (Remarks to the Author):

The authors did an excellent job in answering my concerns and questions.

One thing that I would like to see changed in the revised manuscript is adding "a-priori" in the text in Abstract and Conclusion:

Future research should include trials commencing preconception and provide results disaggregated by A-PRIORI defined participant characteristics including social and environmental factors, clinical traits, and other novel risk factors to predict GDM prevention through interventions.

I find this important, since in my experience, high-quality journals do not accept analyses that have not been defined a-priori in a study protocol. It also prevents "fishing" for significance in a subgroup.

I have no further comments on the revised manuscript.